# Food Front-of-Pack Labelling and the Nutri-Score Nutrition Label—Poland-Wide Cross-Sectional Expert Opinion Study

**DOI:** 10.3390/foods12122346

**Published:** 2023-06-12

**Authors:** Mariusz Panczyk, Hubert Dobrowolski, Beata I. Sińska, Alicja Kucharska, Mariusz Jaworski, Iwona Traczyk

**Affiliations:** 1Department of Education and Research in Health Sciences, Faculty of Health Science, Medical University of Warsaw, 00-581 Warsaw, Poland; mariusz.jaworski@wum.edu.pl; 2Department of Functional and Organic Food, Institute of Human Nutrition Sciences, Warsaw University of Life Sciences, 02-776 Warsaw, Poland; hubert_dobrowolski@sggw.edu.pl; 3Department of Human Nutrition, Faculty of Health Sciences, Medical University of Warsaw, 01-445 Warsaw, Poland; beata.sinska@wum.edu.pl (B.I.S.); alicja.kucharska@wum.edu.pl (A.K.); iwona.traczyk@wum.edu.pl (I.T.)

**Keywords:** consumer awareness, dietary choices, front-of-pack label, Nutri-Score, nutrition labelling

## Abstract

Front-of-pack labelling (FOPL) systems, such as the Nutri-Score, play a crucial role in promoting healthy diets and raising consumer awareness. Our study aimed to gather the opinions of Polish experts on the Nutri-Score and its relation to an ideal information system. We conducted a Poland-wide expert opinion study using a cross-sectional design survey with 75 participants, who had an average of 18 ± 13 years of experience and were mainly employed at medical and agricultural universities. The data were collected with the CAWI method. The results showed that the most important features of an FOPL system were clarity, simplicity, consistency with healthy eating recommendations, and the ability to objectively compare products within the same group. While more than half of the respondents believed that the Nutri-Score provides an overall assessment of a product’s nutritional value and facilitates quick purchasing decisions, it falls short in helping consumers compose a balanced diet and cannot be applied to all product groups. The experts also expressed concerns about the system’s ability to account for a product’s degree of processing, full nutritional value and carbon footprint. In conclusion, Poland’s current labelling system needs expansion, but the Nutri-Score requires significant and detailed changes and validation against national guidelines and expert expectations before implementation.

## 1. Introduction

In recent years, there has been a tendency for the public to lead healthier lifestyles and to change to healthier eating patterns [1]. Increasingly, consumers are looking for products in the shops that not only meet sensory or quality criteria, but also have an adequate nutritional value. Often, however, the mandatory information on the packaging is insufficient for the selection of a valuable product, and its interpretation by people without adequate knowledge may take a long time, often leading to the choice of a product with less beneficial properties [2,3,4]. Appropriate information and labelling on packaging therefore plays a huge role in making more informed dietary choices, and thus in improving the quality of the public’s diet and preventing related diseases [5,6].

One of the important tools to help consumers make healthier food choices is front-of-pack labelling (FOPL). As indicated by the World Health Organization (WHO), FOPL systems are an important tool to promote a healthy diet by raising consumer awareness [7]. From a practical point of view, it is important that a well-designed system simply, quickly and intuitively helps the consumer make the right choice and, above all, does not mislead them about the nutritional value of the product on which it is displayed. Indeed, a meta-analysis by Croker et al. (2020) showed that FOPL systems encourage the purchase and consumption of healthier foods [8]. Feteira-Santos et al. (2020), on the other hand, emphasized that, although FOPLs have a positive impact on dietary choices, there is no clear evidence to conclude that specific knowledge can influence the consumer’s understanding of the nutritional value of a product or the choice of healthier foods [9]. It is therefore important to analyse the FOPLs present in the market in order to minimize the risks to consumers of misunderstanding certain labels and thus making poor nutritional choices.

One food labelling system gaining popularity in recent years is the Nutri-Score system, developed in France. On the basis of the energy, sugars, saturated fat and sodium content, the amount of fruits and vegetables, fibre, and protein [10], as well as, starting from 2021 [11], nuts, legumes and oils (rapeseed, walnut and olive oil), this system provides a score and classifies products into more or less healthy per 100 g/mL of product, individually for beverages, cheese, fats and other types of food products. However, the Nutri-Score system is an across-the-board system and is not a system that is category-based for food products.

The Nutri-Score system has been analysed in many studies. While many of these studies have shown that the system fits in with healthy eating and public health recommendations [12], as well as in helping to make healthier dietary choices and improving the health of the population [12,13,14,15], the scientific community has also highlighted the flaws and systemic shortcomings of this algorithm, or pointed to the lack of clear and reliable evidence of its positive impact on consumers’ dietary choices [16,17,18,19,20].

The European Food Safety Authority (EFSA) issued an opinion in which it recognized the need for harmonized, mandatory, front-of-pack nutritional labelling, which aims to improve the health of the general population by promoting healthier dietary choices, as well as increasing the consumption of products with an adequate nutrient profile [21]. At the same time, however, the WHO stated that, despite the benefits and assistance provided by FOPL, it is unable to recommend any specific system and sees the need for further studies using different labels [7]. Individual countries, based on the opinions of their experts and scientific societies, are therefore introducing FOPLs based on internal regulations in an effort to improve the health of their citizens. For example, Warning Signs have been introduced as mandatory in Chile, Peru, Uruguay, Mexico, Colombia and Argentina, while by 2023, the Nutri-Score was introduced, as recommended, in seven European countries, including France, Spain, Switzerland and Germany, among others [5]. Several countries also officially did not support the introduction of the Nutri-Score system, including Italy, Czech Republic, Cyprus, Greece, Hungary, Latvia and Romania. The implementation of the Nutri-Score system is also being discussed in other countries, including those in the European Union, despite the ambiguities in its benefits.

Given the differences in opinions on the Nutri-Score system, its advantages and disadvantages, and the need for a good FOPL that is understandable to the consumer and promotes health-promoting eating behaviours, the aim of our study was to discover the opinions of Polish nutrition and dietetics experts regarding the Nutri-Score algorithm and labelling in relation to the actual need for the features and functions of an ideal information system. To the best of our knowledge, this study represents the first example in a European country of surveying the opinions of specialists in nutrition and dietetics, who are affiliated with scientific societies, regarding the perception of the Nutri-Score system.

## 2. Materials and Methods

### 2.1. Design

This Poland-wide expert opinion study, using a cross-sectional design, included a sample of 75 specialists in nutrition and dietetics who were members of Polish scientific societies. The study was conducted between March and April 2022. The study employed the definition of an expert opinion as described by Schünemann, et al. [22], as a view or judgment formulated on a particular subject. In this study, the opinions expressed by the experts were not evaluations of existing scientific evidence, but rather based on their individual experiences, perceptions, and knowledge of FOPL.

### 2.2. Sample and Setting

Purposive sampling was used in the recruitment of the study group in order to control the study participants and minimize the risk of participation of those subject to the exclusion criterion, as well as to ensure the participation of specialists who deal with nutrition, dietetics and food labelling issues. In addition, a snowball method of selection was used in which study participants could recruit further participants. The respondents received an individual invitation with a specially generated link, ensuring that only the abovementioned specialists participated in the study.

Experts and specialists in the field of nutrition and dietetics who could take part in the study included members of the Committee for Human Nutrition Sciences of the Polish Academy of Sciences (CHNS PAoS) (*n* = 39), the Polish Society of Dietetics (*n* = 300), the Polish Society of Nutrition Sciences (*n* = 200), as well as the people involved in education in the field or specialisation of dietetics (academic lecturers of 39 public universities and 25 non-public universities (*n* = 640)). In total, it was estimated that there were approximately 1200 eligible professionals in Poland. Based on this data, the minimum number of participants in the study was estimated. Of the 1200 professionals, 110 with scientific and/or educational links to human nutrition and dietetics were sent invitations to participate in the study. Of those invited, 70 professionals attempted to participate in the study, of which 65 fully completed the questionnaire. In addition, through snowball recruitment, 20 people joined the study, of whom 10 completed it. In total, therefore, 75 professionals meeting the inclusion criteria took part in the study (Figure 1).

### 2.3. Instrument

The questionnaire was developed by an expert panel consisting of specialists in dietetics and human nutrition, legal aspects of food labelling, psychology, social research methodology and statistics. The initial version of the questionnaire was reduced by consensus in terms of the number of questions it contained. The questionnaire was then analysed through consultation with external experts, followed by a pilot study to test its usefulness.

The final version of the questionnaire contained five sections: (I) background information on food labelling systems, (II) features of an ideal food labelling system, (III) Nutri-Score’s front-of-pack food labelling system, (IV) implementation of Nutri-Score’s food labelling system in Poland, and (V) selected personal data of respondents. In the first section, the respondents were asked about the role fulfilled by the current food labelling system, what additional labelling elements would help consumers to make the right nutritional choices, the degree of familiarity (on a 5-point scale) with the voluntary on-pack labelling in the market, and the need (on a 5-point scale) to expand the mandatory RWS. In the next section, the respondents indicated (on a 5-point scale) the importance of the features of the ideal system from among the 17 possible product labelling features indicated in the questionnaire, and whether, in their opinion, there was a need for additional front-of-pack labelling. In the third section, the 17 desirable features of FOPL that were presented in the previous section were once again presented, but this time respondents were asked to rate these features in relation to the Nutri-Score system (meets feature; does not meet feature; no opinion). In the fourth section, experts were asked whether the Nutri-Score system should be introduced in their country.

### 2.4. Data Collection

The data were collected with a self-report online survey (computer-assisted web interviewing). The questionnaire was distributed with the aid of the LimeSurvey web platform. The link to the survey was shared with members of the Committee on Human Nutrition of the Polish Academy of Sciences, the Polish Society of Dietetics, the Polish Society of Nutritional Sciences and selected academic lecturers of Polish medical and agricultural universities and researchers from scientific institutions. The digitalized data were secured and archived for five years, in line with the standard operating procedures of the Medical University of Warsaw.

### 2.5. Ethical Considerations

Information about the objectives of the study, the methods of data analysis and archiving were provided in a written form. Moreover, the respondents were assured that their personal data would be used solely for research purposes. Each participant could withdraw from the study at any stage. According to the Local Inspector for Personal Data Protection, considering the type of data collected within the framework of the study, no additional consent for personal information processing had to be sought from the participants. The protocol of the study was approved by the Bioethics Committee of the Medical University of Warsaw (protocol code KB/76/2021 date of approval 15 December 2021). This study complied with the Declaration of Helsinki.

### 2.6. Data Analysis

All calculations were performed using STATISTICATM 13.3 software (TIBCO Software, Palo Alto, CA, USA). The collected data were analysed using descriptive statistics methods. Structural indicators such as number (N) and frequency (%) were determined. The confidence intervals for both the mean and frequency were estimated using the bootstrap method. Specifically, the percentile method was employed, and 1000 bootstrap samples were generated. For selected variables, a statistical description was also generated using measures of central tendency (mean and median) and variability (standard deviation).

## 3. Results

### 3.1. Sample Characteristics

The expert panel included 75 professionals with an average length of experience of 18 ± 13 years (median 15.0 years). The vast majority of study participants had a length of work experience of 25 years or less (*n* = 58; 77%). The main place of work of the surveyed experts as specialists in dietetics and human nutrition was a medical and/or agricultural university (36 and 35%, respectively). More than half of the respondents (55%) described the specificity of their professional activity as teaching work with students on topics related to nutritional labelling. Nearly one-quarter of the respondents (24%) declared that they teach school/workshop classes on nutrition labelling. A summary of the characteristics of the surveyed experts is presented in Table 1.

### 3.2. Front-of-Pack Nutrition Labelling Systems Knowledge and Suggestions for Improvement

Of the front-of-pack food labelling systems on the market, RWS/GDA (median familiarity score 5.0), Multiple Traffic Lights (MTL) (median familiarity score 5.0) and Nutri-Score (median familiarity score 4.0) were the best known. The least familiar were the Keyhole and SENS systems. The principles of the RWS/GDA system were at least somewhat understood by 92% of respondents, the principles of the MTL system by 86.7% and the Nutri-Score system by 74.7%.

Only 38% of experts believed that the current food labelling system is sufficient. Of the possible changes to the labelling system indicated by the experts, most suggestions were related to the need to include the presence of bioactive compounds, vitamins, minerals and trans fatty acids in labelling (*n* = 51.68%). In addition, 4% of respondents (*n* = 3) provided their own suggestions for additional data, such as information that the product consumed in excess may contribute to the development of civilization diseases, highlighting which main ingredients in the product are a source of specific nutrients, or taking into account the percentage of fulfilment of consumption recommendations in relation to proteins, fats (including saturated fatty acids, monounsaturated fatty acids, polyunsaturated fatty acids), monosaccharides, salt and fibre.

### 3.3. Features and Functions of Front-of-Pack Nutrition Labelling System vs. Nutri-Score Labelling System

Table 2 shows the importance of the features that a properly designed FOPL system should have and the fulfilment of these features by the Nutri-Score system. A total of 14 of the 17 evaluated FOPL features and functions received an average importance score of more than 4.0 on a 5-point Likert scale from the expert panel. According to the experts, the most important features of a FOPL system are that it should be clear, understandable and simple (4.92 ± 0.27), consistent with healthy eating recommendations (4.91 ± 0.29), and allow objective comparisons of products from the same product group offered by different manufacturers (4.59 ± 0.50). The least important features, according to the evaluation system, were the consideration of the carbon footprint (3.20 ± 0.97), not depreciating any product group (3.57 ± 1.07) and allowing objective comparisons of different product groups amongst themselves (3.76 ± 1.18).

More than half of the respondents stated that the Nutri-Score system succeeds in features such as providing an adequate overall nutritional assessment (65.3%; ranked 6th in terms of importance out of 17 features, according to the respondents) and allowing for a quick purchasing decision (81.3%; ranked 11th in terms of importance). More than half of the respondents indicated that the Nutri-Score system fails considerably in features such as facilitating the composition of a balanced diet (80.0%; 5th place in terms of importance out of 17 features, according to respondents), feasibility for all product groups (57.3%; ranked 7th in terms of importance), consideration of the degree of processing of the product (76.0%; ranked 8th in terms of importance), consideration of the full nutritional value of the product (74.7%, ranked 12th in terms of importance), not depreciating any product group (64.0%; ranked 16th in terms of importance) and considering the carbon footprint (54.7%; ranked last in terms of importance). The most balanced responses, and therefore divided opinions among the experts, were observed for attributes such as compliance with healthy eating recommendations (36.0% vs. 34.7%, for fulfilled and unfulfilled attribute, respectively), helpfulness in nutrition education (34.7% vs. 28.0%, for fulfilled and unfulfilled attribute, respectively), encouragement of careful reading of the composition and nutritional value of products (29. 3% vs. 38.7%, for fulfilling and not fulfilling the attribute, respectively), universality for all EU countries (32.0% vs. 33.3%, for fulfilling and not fulfilling the attribute, respectively) and not depreciating regional, traditional and organic products (21.3% vs. 30.7%, for fulfilling and not fulfilling the attribute, respectively).

### 3.4. Implementation of the Front-of-Pack Nutrition Labelling System

The majority of survey participants (76%; *n* = 57) saw the need for additional labelling in the form of FOPL, and 65.33% (*n* = 49) of respondents were willing to accept an algorithm-based system that only considers selected nutrients. Forty-eight per cent of survey participants (*n* = 36) believed that the FOPL label has the greatest educational value for the average consumer (with 49.33%; *n* = 37 of respondents believing that the nutrition tables with RWS have the greatest educational value).

With regard to the Nutri-Score system, only 6.67% (*n* = 5) of the experts believed that this system should be introduced as mandatory in its current version. More than half (58.67%; *n* = 44) of the experts believed that the system could be introduced as mandatory, but only in a modified version. The opposite view was held by 24% (*n* = 18) of respondents who believed that the Nutri-Score system should not be introduced as compulsory in Poland at all. Eight respondents (10.67%) had no opinion on the obligatory introduction of this system in the country. The graph in Figure 2 shows the breakdown of the experts’ responses regarding the possibility of an obligatory introduction of a Nutri-Score system in Poland.

## 4. Discussion

This study summarized and systematized the opinions of a representative group of experts in nutrition and dietetics on the Nutri-Score system, its strengths and weaknesses in relation to the desirable features of FOPL, as well as the possibility of introducing this system in Poland, together with an indication of the direction of possible changes in the labelling of food products. According to our knowledge, this is the first study of its kind on FOPL and the Nutri-Score system.

### 4.1. Current Food Labelling and Prospects for Development

The food labelling system in Poland should be expanded and improved. Only 38% of the experts believed that the current food labelling system is fully sufficient. Regulation (EU) No 1169/2011 of the European Parliament and of the Council of 25 October 2011 on the Provision of Food Information to Consumers, which is in force in the European Union, imposes an obligation to provide a wide range of information, such as the name of the food, the list of ingredients or the date of minimum durability or ‘use by’ date, as well as information on the nutritional value of the product. According to the regulations, when declaring the nutritional value, the energy value as well as the amount of fat, saturated fatty acids, carbohydrates, sugars, protein and salt must be declared as mandatory. In addition, the content of the mandatory nutrition declaration may be supplemented with information such as the content of mono- and polyunsaturated fatty acids, polyols, starch, fibre and selected vitamins and minerals listed in the annexes to the regulations. The Regulations also mention exceptions of food products to which these guidelines do not apply [23]. The labelling criteria in the Regulations are not sufficient according to the experts surveyed. The experts see a need to extend the compulsory labelling with additional information, either those listed in the Regulations or not included in the Regulations.

Selecting specific indicators that provide evidence of a product’s nutritional value and that are useful for food profiling is challenging. On the one hand, an excessive amount of component information can overwhelm and confuse consumers and complicate algorithms for profiling and categorising foods, while on the other hand, it is easy to overlook components that should be considered in such procedures. Among the indicators suggested by the experts that could effectively expand the consumer information were mainly information such as the content of bioactive compounds, vitamins, minerals and trans fatty acids. These suggestions are in line with those observed in the literature. As indicated by EFSA (2022) in its opinion on food profiling for the development of mandatory FOPL, the intakes of dietary fibre and potassium are below current dietary recommendations in the majority of European adult populations. Moreover, dietary intakes of iron, calcium, vitamin D, folate and iodine are below current dietary recommendations in specific subgroups of European populations [21]. The inclusion of at least some vitamins and minerals, especially those that are often observed to be insufficiently consumed, seems to be helpful in determining the nutritional value of a food product. Potassium is one of the components responsible for the body’s water and acid–base balance and is also involved in regulating nerve and muscle cell function [24], protecting against hypertension and, perhaps, in improving bone health [25]. An adequate supply of iodine is required for the secretion and function of thyroid hormones, and thus influences cell metabolism and differentiation, and is important for foetal development and gene expression [26]. Iron is crucial in cell metabolism, oxygen transport and enzymatic reactions [27]. Calcium and vitamin D are required for normal growth and development and play important roles in bone health maintenance [28]. In addition, vitamin D, plays a role in affecting cell proliferation and differentiation, and is involved in immune function, inflammation, anti-oxidation and anti-fibrosis, and vitamin D deficiencies, which are observed in a significant part of the population, have been linked to bone metabolic disorders, tumours, cardiovascular diseases, and diabetes [29]. The special role of folate is mainly related to reducing the risk of neural tube defects, but folate nutritional status has also been linked to chronic diseases such as cardiovascular diseases, cancer and cognitive dysfunction [30]. The inclusion of some vitamins and minerals in the construction of an algorithm assessing the nutritional value of a food product, especially those whose inadequate intake is commonly observed in Europe, is therefore necessary for the algorithm to perform its function properly.

Other authors also suggest other components that should be considered when constructing FOPL. The previously mentioned paper by EFSA (2022) mentioned the widespread insufficient intake of fibre [21]. Prieto-Castillo et al. (2015) highlighted the important role of trans fatty acids, and pointed out that they are not included in FOPL [4]. Additionally, Cannoosamy et al. (2014) asserted that information on energy value, trans fatty acids and cholesterol should be included in FOPL [3]. The Whole Grain Initiative, in their statement and open letter to the European Commission and several stakeholders, called for the inclusion of whole grain in the proposed harmonized mandatory front-of-pack nutrition labelling for the EU [31]. Kissock et al. (2021) also showed in their paper that the inclusion of whole-grain products would improve the algorithm (in relation to the Nutri-Score algorithm) and bring it closer to the overall dietary recommendations [32]. Although the respondents in this study, a group of experts and specialists, did not mention these components as potential improvements to the nutritional information system, it is difficult to deny the validity of these suggestions. Dietary fibre has a number of functions, such as maintaining normal bowel function and alleviating constipation, stimulating microbial growth and increasing faecal bulk, and its intake has been inversely associated with the risk of developing CVD and T2DM [21]. The main sources of fibre are whole grain cereals, legumes, fruits, vegetables, and potatoes when eaten with the skin [21], so the inclusion of whole grains seems to be justified, at least in terms of ensuring an adequate fibre supply. Trans fatty acids influence the regulation of physiological processes such as lipid metabolism, inflammation, oxidative stress, endoplasmic reticulum stress, autophagy, and apoptosis and has been linked to cardiovascular disease and ischemic heart disease [33]. Additionally, increased cholesterol intake has been linked to increased total and cardiovascular mortality [34].

There are many potential ways to expand FOPL systems with nutritionally relevant information. Certain information is indicated both by the surveyed experts and the authors of numerous publications and position papers. Referring these elements to the Nutri-Score system, it should be pointed out that this system is based on the content of selected nutrients listed as mandatory in the Regulations, in addition to supplementing the mandatory information with additional information in the form of the amount of fibre, as well as including some additional information in the form of selected products and fats of plant origin [10,11]. It does not, however, include some information which, from a public health point of view, affects the nutritional quality of the product, and which were indicated in the expert opinion and many other studies: the content of selected vitamins and minerals, or the content of trans fatty acids and cholesterol.

### 4.2. Key Features of FOPL vs. Nutri-Score System

In the opinion of the experts participating in the study, the most important features of FOPL are that the labels should be simple, legible, understandable, consistent with nutrition recommendations and allow comparisons of products from the same group. Additionally, while a significant proportion agreed that the Nutri-Score system fulfils these features, this was not the case for half of the specialists, and in terms of compliance with the nutrition recommendations, the answers in the study group were very divergent. The authors of other studies had the opposite view. The study by Hercberg et al. (2021) emphasized that the Nutri-Score system is simple, clear and understandable for consumers, allowing them to make healthier dietary choices. They also showed an improvement in the nutritional quality of the purchases of people guided by the Nutri-Score when choosing products, demonstrating the compatibility of this algorithm with the principles of healthy eating [12]. Similarly, the understanding of the system was demonstrated in the study by Fialon et al. (2021) [35]. However, both these and previous studies were conducted with the participation of the algorithm developers. Compliance with national dietary recommendations was observed, among others, in a Dutch study [36]. Consistency of this system with dietary recommendations has also been shown by other studies [37,38]. A study with Greek consumers also found the system to be clear, visible and easy to understand. However, this study compared this system only with GDA labels [39]. A study in a Slovakian population showed that the Nutri-Score system was effective in comparing products belonging to the same group and performed better than Nutrinform for cereals and bars, but worse for yoghurts [40]. Włodarek and Dobrowolski (2022), on the other hand, showed that the Nutri-Score is unable to distinguish between two packages of certain cereal products, which receive the highest category but differed in values such as glycaemic index, or fish, where fatty fish received a worse score than lean fish despite its higher PUFA content [16]. Thus, there is a lack of independent research indicating the readability, simplicity and understanding of the Nutri-Score by consumers, as well as the translation of this index into dietary health. The results of studies comparing products within the same group were, in turn, inconclusive.

### 4.3. Positive Features of the Nutri-Score System in Light of the Desirable Characteristics of FOPL

More than half of the respondents indicated that the Nutri-Score system is an algorithm that provides an overall assessment of the nutritional value of a product, as well as allowing for a quick purchasing decision. This opinion may be due to the design of the system. Indeed, the Nutri-Score is a FOPL that has a rather simple design. A five-point rating scale from A to E, together with a colour gradation, allows for a simple and quick overall assessment of the product and a purchasing decision. The experts’ opinions on the ability of the algorithm to provide an overall nutritional score is probably due to the components that are used to calculate the overall score. Indeed, a Nutri-Score system rating is mainly based on the components that are declared by the manufacturer on the back of the package with a few additional indicators (fruit content or selected fats). However, this does not prove the effectiveness of this system. The ability to make a quick purchasing decision does not necessarily indicate the consumer’s understanding of the Nutri-Score system and only demonstrates the understanding and distinguishing of the colours used in product assessment. The colours green and red, corresponding to the recognized signals, may be easier to understand and interpret, with green being associated with safety and the ‘start’ signal, and red with danger and the ‘stop’ signal. An overall assessment of the nutritional value of a product can also be disastrous. Such a general assessment does not consider a number of other factors, such as the content of vitamins, minerals, other bioactive components and essential fatty acids, nutritional value of protein or glycaemic index. Each of these characteristics undoubtedly influences the nutritional value and omitting them may lead to an incorrect interpretation of the nutritional value of a given product.

Like the respondents involved in this study, the developers of the system also emphasized that it meets the criteria of enabling a quick purchasing decision and being able to make an overall assessment of the nutritional value of the product. They also highlight other criteria that are required for a good and reliable FOPL system according to the Joint Research Centre [41]. Similarly, Goiana-da-Silva et al. (2021) showed that the Nutri-Score is less misleading and allows a quicker decision compared to RIs. However, on the other hand, RIs in consumer opinion provided more information and were more trusted [42]. A study with 814 consumers from Morocco also pointed to the possibility of making a quick decision with the Nutri-Score system, which was justified by the easy-to-interpret colour scheme [43]. Additionally, Marczuk et al. (2021) found that the Nutri-Score allows a quick comparison of products with each other [44]. The authors of many studies also unanimously point out that the Nutri-Score is a good tool for general nutritional assessment. This is highlighted in the work of Egnell et al. (2020) [45], Hercberg et al. (2022) [12], Ferreiro et al. (2021) [17], and Julia et al. (2021) [46], among others. However, a quick purchasing decision is not necessarily an explicitly positive feature. A quick product choice can be detrimental if products are labelled in a misleading way, as discussed below.

### 4.4. Features of an Ideal FOPL That the Nutri-Score Does Not Meet

The respondents indicated that the Nutri-Score system does not meet features such as making it easier to compose a balanced diet, being able to be used for all product groups, taking into account the degree of processing of the product, taking into account the full nutritional value of the product, not depreciating any product group and taking into account the carbon footprint.

Of these characteristics, facilitating the composition of a balanced diet was one of the most important properties of an ideal FOPL indicated by the experts participating in this study. Many studies have shown that the Nutri-Score system has the ability to indicate a better product within the same category (e.g., comparing several types of pizzas with each other [47]). However, it cannot be clearly indicated that this will translate into an overall dietary pattern. Additionally, as the results of the study presented by Kupirovič et al. (2020) indicated, all tested FOPL systems translated into making healthier dietary choices when they were in line with dietary recommendations and able to distinguish between a healthier and less healthy product [48]. However, the assessment by the FOPL system of single, isolated products makes it impossible to implement the Nutri-Score to improve the overall quality of the diet. Indeed, the consumption of products categorized as the healthiest according to this index alone does not guarantee that nutritional deficiencies will not arise if the diet is not adequately varied [49]. The experts’ opinion is therefore fully justified. Similarly, Carruba et al. (2022) indicated that the Nutri-Score system does not provide any assistance in deciding the overall dietary composition, nor does it facilitate in any way an appropriate combination of various foods [50].

The experts pointed out that the Nutri-Score system is not applicable to all product groups. As mentioned, Nutri-Score relies on individual nutritional elements as components to calculate its score. It is therefore possible to calculate a Nutri-Score in any product where these data are declared by the manufacturer. The legislation mentions a small group of products that are not covered by the mandatory nutrition declaration, which may make assessment with this system more difficult, but not impossible since even such products have a nutritional value. For obvious reasons, Nutri-Score, as one of the front-of-pack labelling systems, will not be present on fresh products that do not have this packaging. This, however, raises the question of whether fresh produce that could be sold unpackaged should undergo an assessment. It does not seem reasonable, especially from an ecological point of view, to sell packaged fruit just for the gain of a scoring system. However, this is a feature common to all FOPL, not just the Nutri-Score system.

The experts participating in this study also stated that the Nutri-Score system does not take into account the degree of processing of the product. This opinion is supported by studies by other authors. As pointed out by Ferreiro et al. (2021), more than half of the products receiving a category from B to D score were ultra-processed foods, while at least 26% of products in all Nutri-Score categories (including category A) could be classified as ultra-processed [17]. Additionally, while there is obviously a large discrepancy in the content of such products between categories A and E, the presence of highly processed foods in such a high percentage raises some concerns. Consumption of highly processed foods promotes cardiovascular diseases [51] and gastrointestinal disorders [52] and increases the risk of mortality [53], among others. However, to our knowledge, the authors of the Nutri-Score system are working on introducing an appropriate warning against ultra-processed foods [54], which can certainly be credited with improving the system and eliminating a sizable systemic error. However, the work on this topic is not yet published in peer-reviewed journals, so it is difficult to conclude unequivocally on the effectiveness of these countermeasures. Further studies on the introduction of the modification will be needed to assess the effectiveness of this change in order to draw clear conclusions on the matter.

A significant proportion of the study participants also indicated that the Nutri-Score system does not take into account the full nutritional value of a product. As previously mentioned, the algorithm takes into account energy content, sugars, saturated fat, fibre, protein, sodium, fruit, vegetable, nuts, legumes and oils: rapeseed, walnut and olive oil when assessing the nutritional value of a product [10,11]. The developers of the system point out, however, that the components included indicate the content of other nutritional values. As they point out, the inclusion of fruits and vegetables in the calculation was shown to be an excellent proxy for the quantity of certain vitamins, such as vitamin C and pro-vitamin A (beta-carotene), and proteins were selected as a proxy for the quantity of minerals and trace elements in food products, such as calcium and iron [12]. However, to our knowledge, there is no evidence to conclude that the components present for the calculation of nutritional value by Nutri-Score convey the full nutritional value of a food product. The opinion of the experts involved in this study therefore appears to be valid. In addition, while one may agree that involving full product data and all nutritional data may over-complicate the algorithm, making it un-calculable, the inclusion of some components may seem reasonable. The inclusion of certain vitamins, minerals, bioactive compounds, trans fatty acids, cholesterol, EFAs, or CLA may result in an evaluation system that is not overloaded with data but can more accurately and efficiently evaluate products for their nutritional value. This leaves some potential for expansion and improvement of the system.

The experts also pointed out that the Nutri-Score system depreciates certain food groups. This finding was also raised in earlier studies by other authors. As indicated by Włodarek and Dobrowolski (2022), the Nutri-Score system depreciates regional products, beverages with naturally occurring sugars (e.g., juices), fish (especially oily marine fish), and may also depreciate organic products [16]. An interesting conclusion was also reached by Braesco et al. (2022). In their research on the labelling of products containing nuts, they noted that, despite the fact that Nutri-Score awarded positive points for the presence of nuts in a product, they could still be rated lower. As they pointed out, these scores were linked to the higher SFA content and higher energy value of these products. As suggested by the authors, these indicators should not be taken into account when evaluating nut-containing products due to their health-promoting properties, including the reduction in the risk of metabolic syndrome while maintaining body weight with regular consumption and their high nutritional value (MUFA, PUFA, B vitamin, mineral, and polyphenol content) [55].

Finally, experts pointed out that the Nutri-Score does not take into account the carbon footprint. The algorithm of this system does not directly address the carbon footprint, but only focuses on nutritional value. The experts therefore rightly pointed out that the algorithm does not directly address this issue. However, it cannot be said that the algorithm is indifferent to the environment. As indicated in the position paper of the EAT-Lancet Commission on healthy diets from sustainable food systems [56], a healthy environmentally friendly diet will contain an adequate caloric supply, be based on plant-based foods, low amounts of foods from animal sources, a predominance of unsaturated fatty acids over SFAs, and contain low amounts of sugars, highly processed foods and refined grains. Indeed, the Nutri-Score algorithm promotes a reduction in energy intake, sugars and SFAs, an increase in the proportion of plant-based foods, and a planned revision of the system [57] includes a reduction in the promotion of protein from meat products. However, there is no direct studies on whether, if, and how, the Nutri-Score system induces more environmentally friendly diets. Despite clear indications that high-scoring products may be environmentally friendly, there is no clear evidence to support this. These ambiguities were further highlighted by the work of Pointke and Pawelzik (2022), where the evaluation of plant-based substitutes for animal products showed that, although meat substitutes had a better Nutri-Score than meat alone, cheese substitutes received a worse score than cheeses made from animal products due to their lower protein content [58]. It is worth pointing out at this stage that other authors have proposed solutions for improving the algorithm with an environmental aspect, and have also proposed additional solutions similar to the Nutri-Score system to take into account the carbon footprint of the product [59,60].

### 4.5. Implementation of the Front-of-Pack Nutrition Labelling System

As the results indicated, the vast majority of Polish experts participating in the study perceived the need for additional labelling in the form of FOPL. This may also be related to the opinion, where almost half of the respondents indicated that FOPL labels have the greatest educational value for consumers, and thus can effectively inform about the nutritional value of a food product. In addition, the indications of the WHO points out that FOPL is a tool to promote healthy diets through facilitating the consumers’ understanding of the nutritional values of the food and making healthier food choices and driving reformulation by the food industry [7]. It is possible that this has had an impact on the opinions of experts from Poland. Additionally, EFSA notes that FOPLs are helping consumers with their food choices [21]. Numerous studies and meta-analyses also confirmed that FOPLs are an effective tool in encouraging healthier food purchases [8,9], facilitating consumers’ understanding of the nutritional value of food [9], and helping consumers to make choices [61]. Additionally, the European Academy of Paediatrics and the European Childhood Obesity Group indicated that FOPLs are a tool for health promotion which increase consumer awareness on the nutritional qualities of packaged foods and purchasing decisions, and called on the authorities of the European Union to introduce mandatory, uniform and understandable FOPLs in the member states as soon as possible [62]. It is no wonder that these systems are so popular and recognized by the scientific community. Therefore, the initiative for mandatory and uniform FOPL throughout the European Union should be supported, as a tool to improve public health and nutritional awareness, as well as to facilitate quick and easy purchasing decisions based on the product’s nutritional value.

In addition, more than half, 65.33% (*n* = 49), of the respondents were willing to accept a graphical label based on an algorithm that only considers selected nutrients. This opens up a number of avenues when it comes to adopting a particular FOPL system, as many labelling systems have a graphic element. However, there is a trade off with a graphical system as it is a simplified representation of nutritional value. Not all elements can be included in the algorithm that will construct the graphic label. Additionally, while the majority of Polish specialists were of the opinion that such an algorithm may only consider selected nutrients, care should be taken to determine which nutrients will be included and which will be omitted from the algorithm. Undoubtedly, the basis for the development of such a label should be the guidelines prepared by scientific experts from EFSA, which have been supported by an in-depth analysis of epidemiological data and a critical appraisal of the scientific evidence [21].

The last issue raised in the survey was the opinion on the validity of introducing the Nutri-Score system in Poland. Of the experts participating in the study, only five believed that this system should be introduced as mandatory in its current version. The experts, as indicated earlier, saw many disadvantages and imperfections associated with current version of this system and see a need to take into account a few additional factors, such as processing degree, carbon footprint, organic origin, content of bioactive substances, and others, mentioned earlier. While there were advantages, such as assessing the overall nutritional value of a product or enabling a quick purchasing decision, the number of negatives outweighs the positive points of this system. In turn, more than half of the experts (58.67%) believed that the system could be made mandatory, but only in a modified version. The developers of the system certainly also see the need for this, as can be seen from the amendments to the system that have already been introduced once, as well as the adaptation of further ones to improve it. An assessment of the changes that are planned for the system is, however, premature; some changes may be removed and some added. Ongoing analysis of current versions of the system will allow future assessments to be made regarding whether the Nutri-Score system is suitable as a recommended labelling system to be introduced in Poland. However, Polish experts overwhelmingly recognized that this can only happen after profound and appropriate modifications that take into account current evidence-based scientific data as well as EFSA guidelines.

Some respondents (*n* = 18) believed that the system should not be introduced at all, and a smaller number of respondents (*n* = 8) had no opinion on the subject. It is difficult to say what motivated respondents to categorically oppose the introduction of this system in Poland. Perhaps the perceived and numerous disadvantages discussed are, in the opinion of this part of the surveyed group, impossible to improve to a degree that would allow proper functioning of the system. It is worth noting that there are also studies that show the system to be ineffective [19], and although the vast majority indicate its usefulness, almost all studies are conducted with the participation of the authors of the system. It should also be pointed out that studies demonstrating the efficacy and performance of the Nutri-Score system are mostly conducted under laboratory conditions. There are not enough studies to simulate real-world conditions and assess how the system will realistically affect consumer behaviours. As highlighted in Braesco and Drewnowski (2023), the research to date provides some, but insufficient, evidence that FOPNLs can lead to meaningful improvements in consumer behaviours and nutritional quality of the packaged food [5]. Caution on the introduction of this system in the country therefore seems highly advisable.

## 5. Limitations

Our study, despite its careful and meticulous preparation, is not free of flaws. First, it should be noted that the study was conducted at the beginning of 2022, when the latest planned revision of the Nutri-Score system was not yet known. To our knowledge, these are changes that could significantly affect the interpretation and assessment of processed products or certain food groups, such as meats or fats. On the other hand, however, it should be noted that, although the amendment has been announced, it has still not been implemented, and updating the packaging of products on shop shelves can take a long time and be a major challenge for food manufacturers. Second, a drawback of the study is the lack of information on why the Nutri-Score system should not be introduced in Poland, or what specific changes would need to be made to the system to make it effective. This would undoubtedly be a great help to the developers of the system in terms of improvements. However, the purpose of our study was primarily to determine the opinions on the current system and how it could be implemented in the country. It therefore did not include guidance and analysis on how the system should work. These questions would therefore be beyond the scope of our paper.

## 6. Conclusions

In conclusion, this study presented the opinions of a representative sample of specialists and experts with many years of experience and who are engaged in scientific and didactic work in the field of dietetics, nutrition and food packaging. In the opinion of the experts, there is a need to expand the system of package labelling in Poland, and beneficial changes could be seen in front-of-pack labelling in a graphic form. As unquestionable advantages of such a system, they point to its simplicity, legibility, comprehensibility, compliance with recommendations for healthy nutrition and the possibility of comparing products from the same group. According to expert opinion, the most important advantages of the Nutri-Score system are its ability to provide an overall nutritional assessment and make a quick purchasing decision. However, the system does not fulfil important features such as facilitating the composition of a balanced diet, the possibility to apply it to all product groups, taking into account the degree of processing of the product, taking into account the full nutritional value of the product, not depreciating any product group and taking into account the carbon footprint.

## Figures and Tables

**Figure 1 foods-12-02346-f001:**
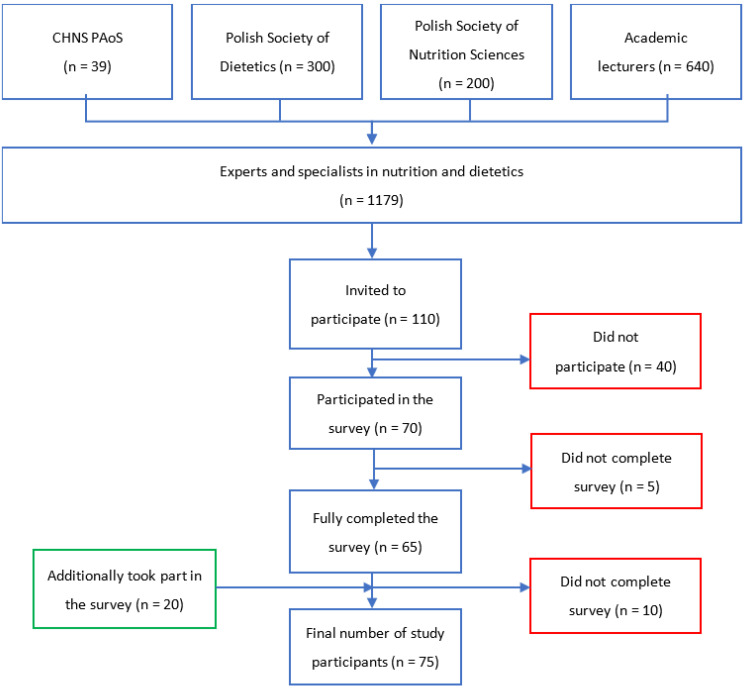
Sampling flow chart.

**Figure 2 foods-12-02346-f002:**
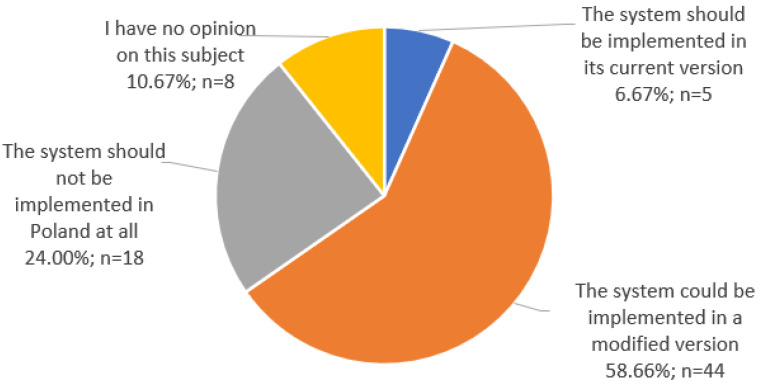
Opinion on the possibility of making the Nutri-Score system mandatory in Poland.

**Table 1 foods-12-02346-t001:** Employment characteristics of study participants (*n* = 75).

	*n*	%
Workplace		
Medical university	27	36
Agricultural university	26	35
Other university	9	12
Scientific research institute	9	12
Hospital/clinic/surgery	12	16
Nature of the work		
Conducting research on nutritional labelling of products	11	15
Conducting classes with students (lectures, seminars, exercises) on nutritional labelling of products	41	55
Conducting workshops/training sessions on nutrition labelling issues	18	24
Providing dietary counselling	26	35

**Table 2 foods-12-02346-t002:** Presence of desirable features/functions of the FOPL system in relation to the Nutri-Score system.

	Newly Designed FOPL		Nutri-Score System		
Feature/Function *	M ± SD (95%C.I.)	Mdn	Meets (%) (95%C.I.)	Does Not Meet (%) (95%C.I.)	No Opinion (%) (95%C.I.)
Clear, understandable, simple	4.92 ± 0.27 (4.86; 4.98)	5	48.0 (37.3; 58.7)	25.3 (16.0; 36.0)	26.7 (17.3; 36.0)
Compliant with healthy eating recommendations	4.91 ± 0.29 (4.84; 4.97)	5	36.0 (25.3; 46.7)	34.7 (25.3; 46.7)	29.3 (20.0; 40.0)
Allows an objective comparison between products from the same product group (e.g., cereals) offered by different manufacturers	4.59 ± 0.50 (4.47; 4.70)	5	46.7 (34.7; 57.3)	28.0 (18.7; 38.7)	25.3 (14.7; 36.0)
Helpful for nutrition education	4.57 ± 0.57 (4.44; 4.71)	5	34.7 (24.0; 45.3)	28.0 (18.7; 38.7)	37.3 (26.7; 49.3)
Facilitates the composition of a balanced diet	4.53 ± 0.55 (4.41; 4.66)	5	5.3 (1.3; 10.7)	80.0 (70.7; 89.3)	14.7 (6.7; 22.7)
Gives an overall assessment of the nutritional value	4.33 ± 0.70 (4.17; 4.50)	4	65.3 (54.7; 76.0)	8.0 (2.7; 13.3)	26.7 (17.3; 37.3)
Applicable to all product groups	4.32 ± 0.70 (4.16; 4.48)	4	2.7 (0.0; 6.7)	57.3 (46.7; 68.0)	40.0 (29.3; 52.0)
Takes into account the degree of processing of the product	4.23 ± 0.86 (4.03; 4.43)	4	1.3 (0.0; 4.0)	76.0 (66.7; 85.3)	22.7 (13.3; 32.0)
Encourages a thorough understanding of the composition and nutritional value of the product	4.20 ± 0.70 (4.04; 4.36)	4	29.3 (18.7; 40.0)	38.7 (26.7; 50.7)	32.0 (20.0; 42.7)
Pushes companies to improve their recipes from a nutritional perspective without having the main objective of obtaining a more favourable rating under the labelling scheme	4.19 ± 0.82 (4.00; 4.37)	4	46.7 (36.0; 58.7)	24.0 (14.7; 33.3)	29.3 (18.7; 40.0)
Allows the customer to make a quick purchasing decision	4.17 ± 0.79 (3.99; 4.36)	4	81.3 (72.0; 90.7)	6.7 (1.3; 13.3)	12.0 (5.3; 20.0)
Takes into account the full nutritional value of the products (macronutrients, minerals, vitamins, bioactive compounds, etc.)	4.08 ± 0.94 (3.86; 4.30)	4	2.7 (0.0; 6.7)	74.7 (64.0; 84.0)	22.7 (13.3; 32.0)
Universal for all EU countries	4.07 ± 0.86 (3.87; 4.26)	4	32.0 (22.7; 44.0)	33.3 (22.7; 44.0)	34.7 (22.7; 45.3)
Does not depreciate regional, traditional and organic products	4.07 ± 0.78 (3.89; 4.25)	4	21.3 (12.0; 32.0)	30.7 (20.0; 41.3)	48.0 (36.0; 60.0)
Allows objective comparisons between different product groups (e.g., a group of sweets to a group of cheeses)	3.76 ± 1.18 (3.49; 4.03)	4	16.0 (8.0; 25.3)	46.7 (36.0; 58.7)	37.3 (26.7; 48.0)
Does not depreciate any product group	3.57 ± 1.07 (3.33; 3.82)	4	4.0 (0.0; 9.3)	64.0 (52.0; 74.7)	32.0 (21.3; 44.0)
Includes the carbon footprint	3.20 ± 0.97 (2.98; 3.42)	3	5.3 (1.3; 10.7)	54.7 (42.7; 66.6)	40.0 (29.3; 52.0)

M—mean, SD—standard deviation, Mdn—median, 95%CI—95% bootstrap confidence interval. * Opinions based on a 5-point Likert scale.

## Data Availability

The datasets generated for this study are available on request to the corresponding author.

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
