# Peer review of "Food Front-of-Pack Labelling and the Nutri-Score Nutrition Label—Poland-Wide Cross-Sectional Expert Opinion Study"

_foods, 2023, doi:10.3390/foods12122346_

Round 1
Reviewer 1 Report
This is a straightforward paper that criticised the Nutri-Score system. It is welll conducted survey among 75 nutritional experts in Poland. I have only a few minor comments:
- as the number of interviewes is just 75, results should be presented only in integers. This applies to many places in the document. Eg in the abstract: 18.1±12.7 it should be 18±13 etc. Table 1 should be: 36, 35, 12, 12, 16 etc. etc.
- experts call on many additional parameters to potentially include in the NS, like processing, carbon footprint, bioactives, organic, local, etc etc. It can be made clear that the original model of NS is not under discussion, and that its current state is asked to the experts.
- authors can make it a bit more clear that NS is an across-the-board system unlike a category-based system.
- around lines 400, it can be made clear that most positive messages stem from the originators of teh NS model.
- The gives a good overview of NS as well as its international reputation and critics. It is a bit long but that is fine with me.
Reviewer 2 Report
Food front-of-pack labelling and the Nutri-Score nutrition 2 label—Poland-wide cross-sectional expert opinion study
Title: The numeral 2 was used instead of the word spelled out. Consider spelling out the number.
General considerations
The study summarized and systematized the opinions of a representative group of experts in nutrition and dietetics on the Nutri-Score system. The experts evaluated the desirable features of food products' front-of-pack labeling, strengths, and weaknesses. The authors considered the possibility of introducing this system in Poland, together with an indication of the direction of possible changes in the labelling of food products. In the opinion of the experts, there is a need to expand the system of package labelling in Poland, and beneficial changes could be seen in front-of-pack labelling in a graphic form. According to the authors' knowledge, it was the first study on front-of-pack food labelling and the Nutri Score system.
Specific comments
I suggested not to simplify the FoLP, with minimal letters. All letters in an acronym must be capitalized. Besides this, after the first time the acronym was used, every other time it is cited, it must be in an acronym, for example, in line 250.
The paper comments about including dietary fiber content, but the author commented nothing about soluble and insoluble fiber content. In some countries, the regulations obligate the food industry to include this information in the nutritional table. In addition to the difference in the physiological performance of the different kinds of fibre, soluble fiber offers two kcal/g, which influences the energy value of the food.
Abstract:
In the abstract, the survey's data collection methodology lacked a short description.
Line 113 to 114. Figure 1: improve the visibility of the data in this Figure.
Line 246. In the title of Figure 2, put the "n" studied. Every Figure and Table must be self-explanatory, and the value of n is always essential information, which must be available without having to resort to the text to retrieve the information.
Line 309. closing parenthesis is missing.
In the non-publishable material.
Only the first page of the one sent to the research volunteers was presented. It would be interesting to show what was experienced in full.
Reviewer 3 Report
This article is well written on (Food front-of-pack labeling and the Nutri-Score nutrition 2 label—Poland-wide cross-sectional expert opinion study), however here are a few suggestions that will improve the quality of the manuscript if followed by the authors
1. The abstract is fine but it needs to be a bit more focused on methodology and results sentences, the conclusion part of the abstract is well written.
2. Key-words in alphabetical order
3. Keep the introduction with recent supportive findings, if you can find some relevant intro of the main title recently published in 2022, that could be much better.
4. Can you write a para in the introduction part about the novelty of this project to grab the reader’s attention?
5. Please mention the full form first and then abbreviations in the following text of the manuscript.
6. Method: Mention the parent article reference that you will be following in your research project
7. What were the criteria for including or excluding relevant scientific work referred in this study?
8. In your research project’s methodology part, it has mentioned that according to inclusion and exclusion criteria…What were the inclusion and exclusion criteria of the study subjects?
9. In Line 310, Cannoosamy et al. (2014 as asserted….. cite references carefully in the manuscript.
10. In the manuscript, add references according to the journal reference style.
11. Limitation
Language and quality of article is poor it must be improved in revised article
